# Clinical and Radiologic Features Together Better Predict Lung Nodule Malignancy in Patients with Soft-Tissue Sarcoma

**DOI:** 10.3390/jcm9041209

**Published:** 2020-04-23

**Authors:** Cecilia Tetta, Antonio Giugliano, Laura Tonetti, Michele Rocca, Alessandra Longhi, Francesco Londero, Gianmarco Parise, Orlando Parise, Linda Renata Micali, Mark La Meir, Jos G. Maessen, Sandro Gelsomino

**Affiliations:** 1Diagnostic and Interventional Radiology, IRCCS Istituto Ortopedico Rizzoli, 40121 Bologna, Italy; cecilia.tetta@ior.it (C.T.); antoniogiugliano986@gmail.com (A.G.); lauratonetti.md@gmail.com (L.T.); 2Unit of general Surgery, IRCCS Istituto Ortopedico Rizzoli, 40121 Bologna, Italy; michele.rocca@ior.it; 3Chemotherapy Unit, IRCCS Istituto Ortopedico Rizzoli, 40121 Bologna, Italy; alessandra.longhi@ior.it; 4Cardiovascular Research Institute Maastricht—CARIM, Maastricht University Medical Center, 6229 ER Maastricht, The Netherlands; francesco_londero@yahoo.it (F.L.); g.parise@maastrichtuniversity.nl (G.P.); o.parise@icloud.com (O.P.); l.micali@maastrichtuniversity.nl (L.R.M.); j.g.maessen@icloud.com (J.G.M.); 5Cardiothoracic Surgery, University Hospital Brussels, 1099 Jette, Belgium; lameir@yahoo.com

**Keywords:** soft tissue sarcoma, pulmonary nodules, metastases, computed tomography scan, lung metastasectomy

## Abstract

We test the hypothesis that a model including clinical and computed tomography (CT) features may allow discrimination between benign and malignant lung nodules in patients with soft-tissue sarcoma (STS). Seventy-one patients with STS undergoing their first lung metastasectomy were examined. The performance of multiple logistic regression models including CT features alone, clinical features alone, and combined features, was tested to evaluate the best model in discriminating malignant from benign nodules. The likelihood of malignancy increased by more than 11, 2, 6 and 7 fold, respectively, when histological synovial sarcoma sub-type was associated with the following CT nodule features: size ≥ 5.6 mm, well defined margins, increased size from baseline CT, and new onset at preoperative CT. Likewise, in the case of grade III primary tumor, the odds ratio (OR) increased by more than 17 times when the diameter of pulmonary nodules (PNs) was >5.6 mm, more than 13 times with well-defined margins, more than 7 times with PNs increased from baseline CT, and more than 20 times when there were new-onset nodules. Finally, when CT nodule was ≥5.6 in size, it had well-defined margins, it increased in size from baseline CT, and when new onset nodules at preoperative CT were concomitant to residual primary tumor R2, the risk of malignancy increased by more than 10, 6, 25 and 28 times, respectively. The combination of clinical and CT features has the highest predictive value for detecting the malignancy of pulmonary nodules in patients with soft tissue sarcoma, allowing early detection of nodule malignancy and treatment options.

## 1. Introduction

Soft-tissue sarcoma (STS) is a heterogeneous group of rare solid tumors of mesenchymal origin that account for approximately 1% of adult malignancies [1]. It includes different types of malignant diseases that can grow in soft tissues such as fat, muscles, nerves, fibrous tissues, blood vessels, or deep skin tissues [2].

A validated model has been proposed to accurately predict the outcome of surgical resection in patients with STS [3]. In contrast, despite the fact that approximately 20% of the patients affected by STS may develop lung metastasis (MTS) [4], an optimal method of screening for lung MTS from STS is not yet available. Furthermore, as far as we know, only a few attempts have been made to detect malignant lung nodules using computed tomography (CT) characteristics but none by clinical criteria [5,6,7].

We test the hypothesis that a model including clinical and CT features may allow discrimination between benign and malignant lung nodules in patients with STS.

## 2. Material and Methods

### 2.1. Ethical Issues

This study was approved by the Institutional Review Board (N. 2018/0012891), which waived the requirement of patient informed consent for this retrospective study according to National Laws regulating observational retrospective studies (law nr.11960, released on 13/07/2004).

### 2.2. Patient Population

The patient population consisted of 170 patients undergoing lung metastasectomy (LMTS) for a pulmonary nodule (PN) at a tertiary STS referral center (Rizzoli Orthopedic Hospital, Bologna, Italy). Data at the time of their first metastasectomy for one or more synchronous or metachronous PNs were collected and examined.

The inclusion criteria were as follows: (1) Availability of histological examination for all resected nodules; (2) Availability of clinical data related to the primary tumor; (3) Available images from 8- or 16-detector row CT scans 3–6 months preoperatively (baseline CT) and up to 7 days from the date of the operation (preoperative CT scan); (4) Available images from postoperative CT scan; (5) Uniform CT protocol with examinations performed with a layer thickness <5 mm. 

The exclusion criteria were: (1) presence of artifacts interfering with diagnosis and masking morphological and densitometric features of nodules, including either patient-based or physics-based artifacts; (2) non-availability of histological examination; (3) non-availability of preoperative and/or postoperative CT scans 

The final study population included 71 patients (Figure 1). 

Clinical data were recorded by one radiologist (A.G.), one clinician (M.R.) and one oncologist (A.L.). 

### 2.3. Definitions and Classifications

All the histologic examinations were reviewed by a pathologist and the grading system was updated following the French Federation of Cancer Centers Sarcoma Group classification [8]. Moreover, STSs were classified following the World Health Organization guidelines updated in 2013 [9], while CT scan features were defined following the Fleischner Society Glossary of Terms for Thoracic Imaging [10]. The recommendations of the American Joint Committee on Cancer were followed for residual tumor R classification [11].

### 2.4. Computed Tomography (CT) Readings

Each of two experienced radiologists (C.T. and L.T.) performed a double-blind revision of all CT scans twice to allow calculation of the intraclass and interclass correlation coefficients. Any disagreements were resolved by consensus reading.

Nodules were classified according to their size, shape, margins, density, and localization based on visual analysis. The shape was classified as round, elongated (length >1.5 width), spiculated, complex, atypical (stria or consolidation), or cavitated. According to their density, nodules were classified as solid (soft tissue density), ground-glass (inferior to soft tissue density), mixed (with both solid and ground-glass components) or calcified (with intralesional calcifications independent of their distribution). Margins were considered well defined or ill defined. Based on the location, nodules were classified as pleural (in contact with pleura), subpleural (within <1 cm from the pleura), parenchymal (>1 cm from pleura but not in contact with the hilum), and hilar (in contact with the hilum). In addition, compared to the baseline CT, any change in growth (decrease, unchanged, increase, and new onset) and density (decrease, unchanged, and increase) were measured. 

### 2.5. Statistical Analysis

Data were tested for normality using the Shapiro–Wilk test and expressed as the mean ± SD or as the median and interquartile range. Statistical differences between malignant and benign PNs for size, primary tumor size, and age were analyzed using the Mann–Whitney test, whereas the other variables were analyzed with Pearson’s Χ^2^ test and Fisher’s exact test. CT findings were evaluated using univariate analysis with generalized estimating equations to consider the clustering effects of multiple nodules per individual.

To identify variables that could be used in predicting malignant PNs and to adjust the effect of the correlation from multiple nodules per individual, logistic regression analysis with generalized estimating equations was performed and weighted by groups’ size. Clinical variables (age, sex, diameter, grading, histology and site of the primary tumor, neo-adjuvant and adjuvant chemotherapy, neo-adjuvant and adjuvant radiotherapy, disease free interval, other sites metastases, histotype, grading surgical margins, local recurrence, and type of surgery) and CT nodule features (size, shape, density, margins, side, lung distribution, change in dimensions, and density between baseline and preoperative CT) were introduced into the analysis. Cluster analysis was used to reduce the number of variables [12,13]. Dummy coding was used for categorical variables with more than one level. Backward stepwise selection was used with iterative entry of variables on the basis of test results (*p* ≤ 0.05). Likelihood ratio statistics with a probability of 0.10 were used for the removal of variables. Three different models were built including CT features alone, clinical features alone, and combined features, respectively, and the C statistic was performed to evaluate their performance [14,15]. The bootstrap method (1000 iterations) was used for validating the results. We tested for interaction terms, and a subgroup analysis was performed to analyze the interactions between the main predictors. The intra- and inter-observer variability was determined by calculating the intraclass correlation coefficient and their 95% confidence intervals. R software v. 3.5.3 (R Foundation for Statistical Computing, Vienna, Austria) was employed for analysis with the packages Desk Tool, Boot, ClustOfVar, ROCR, pROC, and irr. A p value of less than 05 was considered significant.

## 3. Results

### 3.1. Clinical and CT Findings

Clinical characteristics are shown in Table 1. Synovial sarcoma was the most frequent STS histotype (18.3%) and the lower extremities were the most common location (>70%). At surgery, wedge resection was the most frequently employed technique (88.7%) and was performed through open surgery (98.6%). Finally, in 14 patients (19.7%), LMTS was performed in two stages. Forty-six patients (65%) underwent neoadjuvant and/or adjuvant chemotherapy; among them, 40 (87%) received Adriamycin and Ifosfamide.

A total of 160 lesions were examined: 139 malignant and 21 benign (Table 2). The distribution of nodules among patients is schematized in Figure 2. All patients had at least 1 malignant nodule and, among them, 12 showed at least one benign lesion. The histology of benign nodules revealed 15 anthracotic lesions, whereas 6 were intraparenchymal pulmonary lymph nodes. The mean number of malignant nodules per patient was 3.2 ± 1.8, whereas it was 1.7 ± 1.8 for benign lesions. Malignant nodules ranged from 2 to 30 mm, and the median size was 7.6 mm (IQR 4.4–13.0) [3]. In benign nodules the dimension ranged from 2.5 to 9 mm (median 3.5 mm (3.0–5.5), *p* < 0.001) and the majority of malignant lesions had well-defined margins (*p* = 0.017 vs. benign). Compared to the first CT scan, malignant nodules grew significantly while benign lesions remained unchanged (*p* < 0.001). The intra- and inter-observer variability was very low for all the readouts (Appendix A, Table A1).

### 3.2. Multivariable Model Performance

When only clinical covariates were employed, the area under curve (AUC) was 0.70 (95% confidence interval (CI) 0.58–0.81) while when just CT features were used (Figure 3A) the AUC was 0.75 (0.70–-0.79). Additionally, when clinical and CT features were included together, the AUC was 0.92 (95% CI.0.85–0.99, *p* < 0.001 vs. clinical and vs. CT single models).

### 3.3. Results of Multiple Regression Analysis

Among clinical features (Table 3), histological synovial sarcoma sub-type increased the risk of MTS more than 6 times (*p* < 0.001). Similarly, tumor grade III and grade II increased the risk of metastasis more than 4 and 10 times, respectively (both, *p* < 0.001). Finally, with surgical margins R1 and R2 there was a more than 7- and 14-fold increase in the likelihood of malignancy. Among CT features, nodule size resulted in being a significant predictor of malignancy (*p* < 0.001). The calculated cut-off was ≥5.6 mm (Figure 3B). More in detail, any increase of 5 mm nodule size raised the risk of MTS more than 9-fold. Moreover, well-defined margins increased the risk of malignancy almost 1.2 fold, an increased PN size from baseline CT more than 2.3-fold and a new-onset nodule more than 4 fold. Table 3 also reports the results of single regression analyses.

### 3.4. Interaction between Clinical and CT Features

The interaction analysis showed that the likelihood of malignancy increased by more than 11, 2, 6 and 7 fold, respectively (Figure 4A) when histological synovial sarcoma sub-type was associated to the following CT nodule features: size ≥5.6 mm, well defined margins, increased size from baseline CT, and new onset at preoperative CT. Likewise, in case of grade III primary tumor, the OR increased by more than 17 times when the diameter of PNs was >5.6 mm, more than 13 times with well-defined margins, more than 7 times with PNs increased from baseline CT, and more than 20 times when there were new-onset nodules (Figure 4B). Finally, when CT nodule was ≥5.6 in size, it had well-defined margins, it increased in size from baseline CT, and when new onset nodules at preoperative CT were concomitant to residual primary tumor R2, the risk of malignancy increased by more than 10, 6, 25 and 28 times, respectively (Figure 4C). Figure 5 shows some key-CT findings.

## 4. Discussion

The key message of our study is that neither clinical data nor CT findings alone are able to accurately predict malignant nodules. In contrast, when clinical features are supplemented with CT findings, the predictive value increases significantly, with high sensitivity and specificity.

The primary tumor can be cured with wide surgical resection [16] and a reliable model accurately predicting outcome with different treatment modalities is available [3]. In contrast, the discrimination between benign and malignant PNs, particularly in small nodules, is challenging [17,18]. Early detection of pulmonary MTS in these patients is critical for choosing the best treatment option (surgical resection, chemotherapy, radiotherapy, etc.) that significantly influences prognosis [19,20].

CT imaging represents the gold standard for detecting PNs [21,22]. With the introduction of last-generation CT scanners, the number of detected nodules has dramatically increased [23]. As a consequence, a non-negligible number of suspicious lung MTS may be negative for malignancy on histological examination [24]. On the other hand, false-negative CT findings are not infrequent with nodules identified at lung surgical palpation [25]. Therefore, CT screening of suspicious nodules is still an area of active research [5,26,27]. Furthermore, to the best of our knowledge, no study has yet been published that addresses the predictive value of clinical features on lung STS metastases.

We analyzed data from a selected population of patients at the time of their first metastasectomy for one or more synchronous or metachronous PNs. The study population was uniform since only patients with available histological examination were included. Furthermore, as the study was conducted over a 15-year period with different generations of CT scanners. Moreover, only uniform CT protocols were included to avoid potentially unreliable readings. Overall, 160 lesions were identified in 71 patients on CT and 86.9% were metastases.

When a grade III primary STS with a 10 odds ratio (OR) of nodule malignancy has nodule size >5.6 mm or well-defined margins at CT scan, the risk of MTS increases by > 17 and > 13 times, respectively. The cutoff of 5.6 mm confirms previous studies by Nakamura et al. [28] and Rissing et al. [27,29]. This was also shown by Dudeck et al. [5] who demonstrated that larger lung PNs in STS patients have high odds of being malignant.

Our patients had a baseline CT that allowed close follow-up of all nodules. In our experience, grade III associated with increased PN size from baseline CT, raised the risk of malignancy by >7 times which became >20 times higher in the case of the nodule being of new onset. When the latter CT findings were concomitant to R2 residual tumor (OR = 14.2), the risk of MTS increased by more than 25 and 28 times, respectively. Moreover, among STS histological sub-types, synovial sarcoma had the highest probability of malignant nodules (OR = 6.47). This probability increased by more than 11 times when nodule size was ≥5.6 mm. In addition this increment was about by 3 times in nodules with well-defined margins, by more than times when PN size is larger compared to baseline CT, and by more than 7.7 times if there are new-onset nodules.

Notably, our data show that clinical context should not be overlooked in determining the probability of malignancy. Indeed, without the support of clinical data many CT features are not specific; for instance, we found that 30% of malignant nodules did not increase in size and 52.4% of them showed ill-defined margins. In addition, to further complicate the matter, a well-defined nodule is also typical of a benign lesion [30] and the other significant CT features are shared with many other neoplasms [10].

However, our data confirm the irrefutable evidence that thin-section CT scan still play a central role in early detection of PNs [31] but it must be implemented by patient-related clinical factors.

The question arises as what to do when detecting very small nodules at CT scan that do not increase in size and have no defined margins. In the absence of significant clinical factors, a strict follow-up CT scan approach can be reasonable. In contrast, in the case of STS grade ≥II, residual primary tumor ≥R1, and when the primary tumor is a leiomyosarcoma, referral to surgery is, in our opinion, mandatory.

Finally, our data strongly support a multidisciplinary STS team involving radiologists, radiotherapists, oncologists, surgeons, and pathologists for clinical decision-making bearing in mind that professional reduced awareness of STSs as well as the non-specific nature of many symptoms of these rare neoplasms may lead to delays in diagnosis [32]. A prompt detection of PN malignancy is crucial in these patients, considering the highly aggressive nature of STSs, the high risk of MTS outside the lungs [19], and the high success rate gained when MTS is localized only in the lungs [4].

## 5. Limitations

Our study has some limitations that should be pointed out. The first is its retrospective nature which might carry selection and misclassification biases. The second is the small number of patients enrolled and together with the small number of benign lesions. This can be justified by the exclusion of patients from our earlier experience for whom, very often, the histology was not available and CT protocols were different. Third, we did not consider the overall number of lung nodules in our model. Fourth, some of the early CT images were scanned at 3–3.75 mm slice thickness. This could have contributed to partial volume artifact and misclassification of nodule size or density.

## 6. Conclusions

The combination of clinical and CT features has high predictive value for detecting malignancy of pulmonary nodules in patients with soft tissue sarcoma. In the case of synovial sarcoma ≥II, and resection margins ≥R1, the association of PNs >5.6 mm, well-defined margin nodules, increased PN size from baseline CT, and a new onset nodule, increases the risk of malignancy up to 28 times. Further larger studies are necessary to confirm our findings.

## Figures and Tables

**Figure 1 jcm-09-01209-f001:**
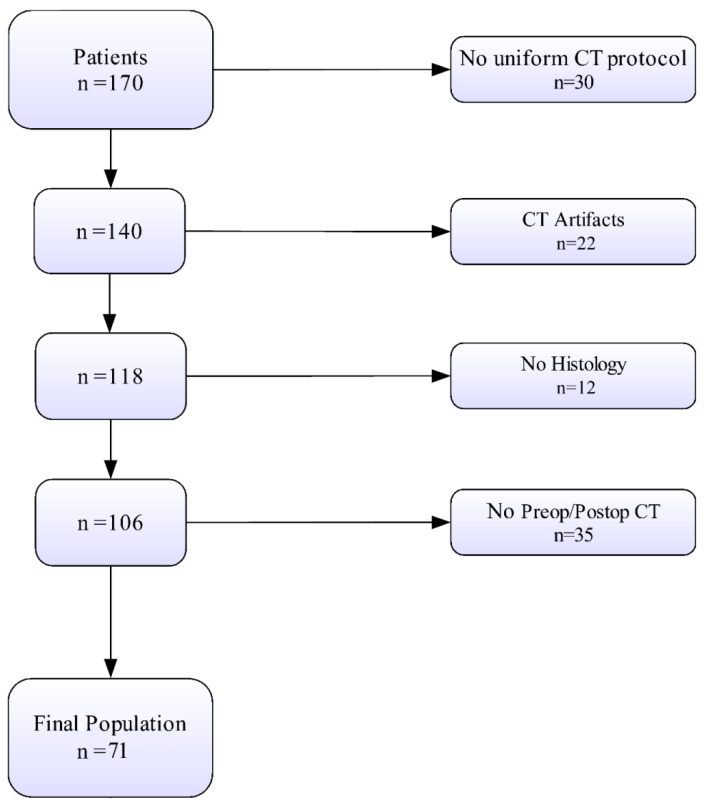
Patient selection. CT, computed tomography

**Figure 2 jcm-09-01209-f002:**
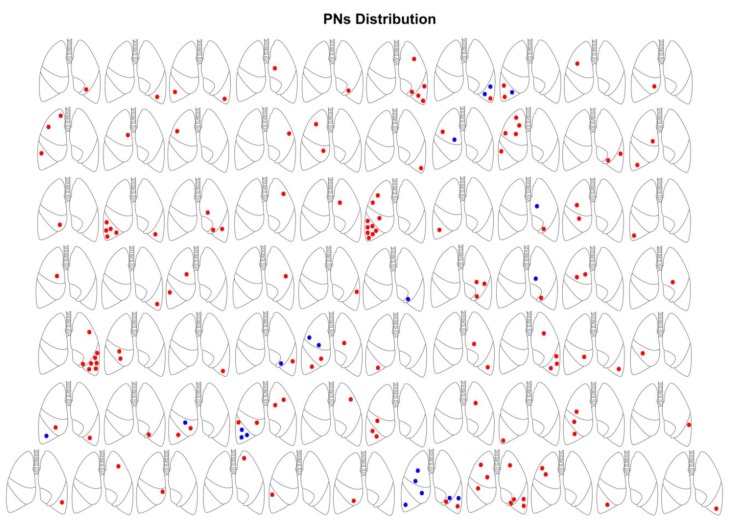
PNs, pulmonary nodules. Schematic distribution of pulmonary nodules among patients. Red nodules were malignant, blue benign at histological examination.

**Figure 3 jcm-09-01209-f003:**
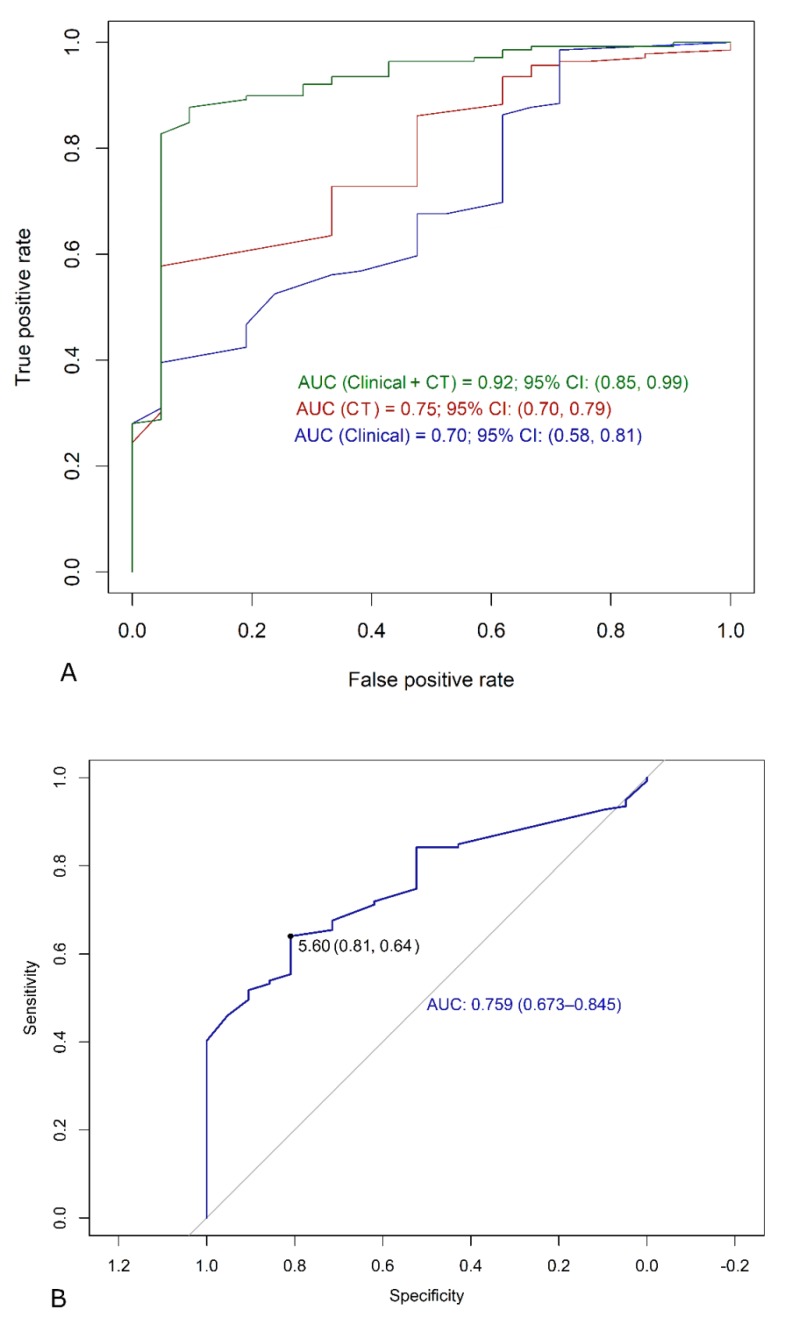
(**A**) Graph shows the results of C statistical analysis of multiple logistic regression models in discriminating malignant from benign nodules. The highest area under the curve (AUC) was achieved with the combination of both clinical and CT predictors (AUC = 0.92), which was significantly higher than that of either clinical (AUC = 0.70) or CT (AUC = 0.81, both *p* < 0.05) alone. (**B**) Cut off calculations for nodule size.

**Figure 4 jcm-09-01209-f004:**
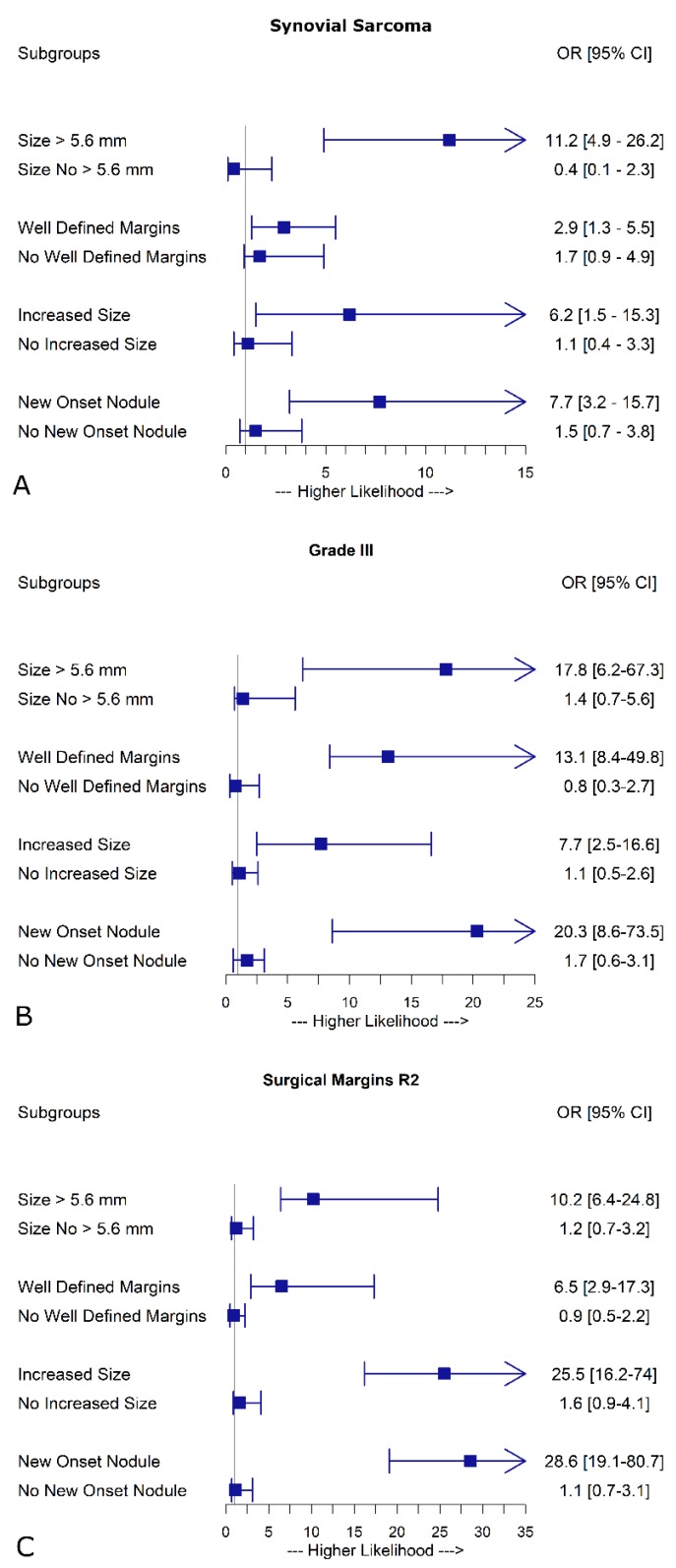
Interactions between multiple metastases and potential influencing factors. (**A**) Interaction between leiomyosarcoma and significant CT features. (**B**) Interaction between primary tumor grade III and significant CT features (**C**) Interaction between R2 surgical resection margins and significant CT features.

**Figure 5 jcm-09-01209-f005:**
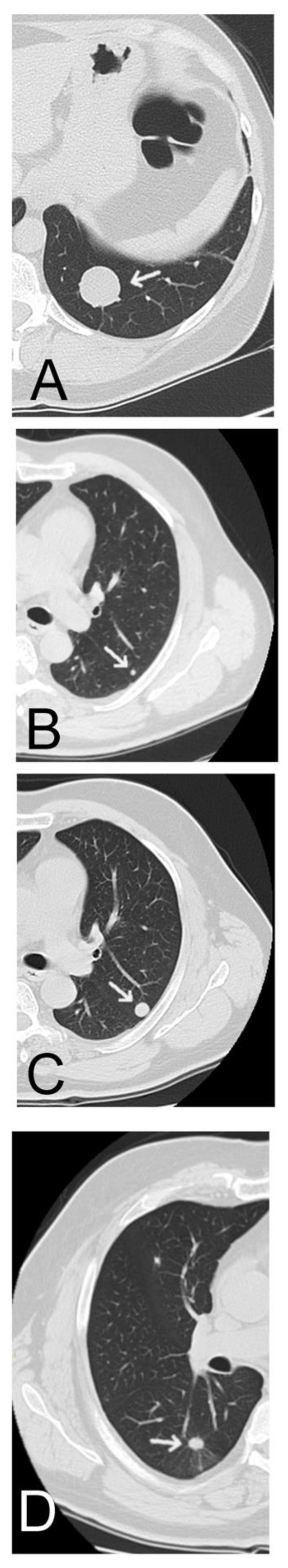
(**A**) The scan shows a malignant 22 mm round-shaped nodule with well-defined margins (arrow) in the left lower lobe in a 44-year-old female patient with synovial sarcoma. (**B**) The scan shows a 3 mm subpleural round nodule with well-defined margins (arrow) at baseline CT. (**C**) Preoperative CT scan obtained approximately 3 months later in the same patients showing a significant increase in size (11 mm, arrow). The patient received no chemotherapy between the scans. (**D)** The scan shows a 10 mm malignant parenchymal nodule with an irregular shape and spiculation (arrow) in the right lower lobe in a 52-year-old male patient with undifferentiated spindle-cell sarcoma.

**Table 1 jcm-09-01209-t001:** Patients and primary tumor characteristics (*n* = 71).

Age		50.00 (42.50, 61.50)
Female sex		25 (35.2)
* Chemotherapy		45 (63.4)
* Radiotherapy		56 (78.9)
Primary Tumor		
Size (mm)		9.00 (6.00, 12.25)
Histology		
	Synovial Sarcoma	13 (18.3)
	Undifferentiated Pleiomorphic Sarcoma	6 (8.5)
	Myxofibrosarcoma	4 (5.6)
	Extra skeletal Myxoid Chondrosarcoma	5 (7.0)
	Epithelioid Sarcoma	3 (4.2)
	Leiomyosarcoma	3 (4.2)
	Extra Skeletal Ewing Sarcoma	3 (4.2)
	Solitary Fibrous Tumor	1 (1.4)
	Epithelioid Hemangioendothelioma	1 (1.4)
	MPSNT	3 (4.2)
	Adult Fibrosarcoma	1 (1.4)
	Dedifferentiated Liposarcoma	2 (2.8)
	Pleomorphic Liposarcoma	1 (1.4)
	Clear Cell Sarcoma of the soft tissue	3 (4.2)
	Dermatofibrosarcoma Protuberans	1 (1.4)
	Undifferentiated Epithelioid Sarcoma	2 (2.8)
	Myxoid Liposarcoma	2 (2.8)
	Undifferentiated Sarcoma	1 (1.4)
	Undifferentiated Spindle-cell Sarcoma	15 (21.1)
	Pleiomorphic Rhabdomyosarcoma	1 (1.4)
Depth		
	Deep	47 (66.1)
	Superficial	16 (22.6)
	Mixed	8 (11.3)
Site		
	Lower limb	39 (54.9)
	Upper limb	20 (28.2)
	Abdominal wall	1 (1.4)
	Back	6 (8.5)

Surgery	R0	52 (73.2)
	R1–R2	19 (26.8)
MTS Surgery	Neck	1 (1.4)
	Gluteus	3 (4.2)
	Pelvis	1 (1.4)

	Wedge Resection	63 (88.7)
	Segmentectomy	3 (4.2)
	Lobectomy	1 (1.4)
	Wedge + Segmentectomy	2 (2.8)
	Wedge + Lobectomy	2 (2.8)
	Open Surgery	70 (98.6)
	VATS	1 (1.4)
	Two-stage MTS	14 (19.7)

Data is shown as numbers (%). Abbreviations. MPNST: malignant peripheral nerve sheath tumor; VATS: video-assisted thoracoscopy; MTS = metastasectomy. * Before or/and after primary tumor resection.

**Table 2 jcm-09-01209-t002:** Pulmonary nodule characteristics.

	Overall	Malignant	Benign	*p*
**Lesions (*n*)**	160	139	21	
**Size (mm)**	6.50 (4.0–12.0)	7.6 (4.4–13.0)	3.5 (3.0–5.5)	<0.001 ^†^
0–5 mm	59 (36.9)	44 (31.7)	15 (71.4)	
5.1–10 mm	52 (32.5)	46 (33.1)	6 (28.6)	0.002 ^‡^
10.1–20 mm	34 (21.2)	34 (24.5)	0 (0.0)	
>20 mm	15 (9.4)	15 (10.8)	0 (0.0)	
**Shape**				
Round	105 (65.6)	98 (69.1)	9 (42.9)	
Elongated	25 (15.6)	18 (12.9)	7 (33.3)	
Complex	19 (11.9)	15 (10.8)	4 (19.0)	0.09 ^‡^
Spiculated	10 (6.2)	9 (6.5)	1 (4.8)	
Atypical	0 (0.0)	0 (0.0)	0 (0.0)	
Cavitated	1 (0.6)	1 (0.7)	0 (0.0)	
**Density**				
Solid	95 (59.4)	84 (60.4)	11 (52.4)	
Ground-glass	33 (20.6)	26 (18.7)	7 (33.3)	0.325 ^‡^
Mixed	28 (17.5)	26 (18.7)	2 (9.5)	
Calcified	4 (2.5)	3 (2.2)	1 (4.8)	
**Margins**				
Well defined	115 (71.9)	105 (75.5)	10 (47.6)	0.017 ^§^
Ill defined	45 (28.1)	34 (24.5)	11 (52.4)	
**Side**				
Right	84 (52.5)	71 (51.1)	13 (61.9)	0.489 ^§^
Left	76 (47.5)	68 (48.9)	8 (38.1)	
**Lobe**				
Upper	52 (32.5)	45 (32.4)	7 (33.3)	
Middle	36 (22.5)	31 (22.3)	5 (23.8)	>9 ^‡^
Lower	72 (45.0)	63 (45.3)	9 (42.9)	
**Location**				
Pleural	26 (16.2)	22 (15.8)	4 (19.0)	
Subpleural	91 (56.9)	81 (58.3)	10 (47.6)	<658 ^‡^
Parenchymal	40 (25.0)	33 (23.7)	7 (33.3)	
Hilar	3 (1.9)	3 (2.2)	0 (0.0)	
**^*^ Size vs. previous CT**				
Reduction	9 (5.6)	9 (6.5)	0 (0.0)	
Unvaried	53 (33.1)	36 (25.9)	17 (81.0)	<0.001 ^‡^
Increase	68 (42.5)	66 (47.5)	2 (9.5)	
New Onset	30 (18.8)	28 (20.1)	2 (9.5)	
**^*^ Density vs. previous CT**				
Reduction	12 (7.5)	12 (8.6)	0 (0.0)	
Unvaried	133 (83.1)	112 (80.6)	21 (100.0)	0.086 ^‡^
Increase	15 (9.4)	15 (10.8)	0 (0.0)	
**^*^ Chemotherapy**				
No	109 (68.1)	95 (74.2)	14 (77.8)	>9 ^§^
Yes	37 (25.3)	33 (25.8)	4 (22.2)	

Variables were expressed as the median [Interquartile Range] or number (%). * between preoperative computed tomography (CT) and previous CT. ^†^ Calculated with the Mann–Whitney test; ^‡^ Calculated with the X^2^ test; ^§^ Calculated with Fisher’s exact test.

**Table 3 jcm-09-01209-t003:** Results of logistic regression analysis in discriminating histologic malignancy.

	OR	95% CI	*p*
**Multiple analysis**			
Clinical Features			
Synovial Sarcoma	6.47	6.18–31.5	<0.001
Grade II	4.76	3.2–36.6	<0.001
Grade III	10.04	8.3–51.1	<0.001
Surgical margins R1	7.6	2.4–20.9	<0.001
Surgical Margins R2	14.2	4.8-80.6	<0.001
CT Features			
Size	9.22	2.97–43.96	<0.001
Well-defined Margins	1.23	1.05–2.91	0.03
Size vs. Baseline CT	2.33	1.19–5.54	0.002
New Onset Nodule	4.65	1.26–13.5	0.03
**Single analysis (Clinical)**			
Leiomyosarcoma	2.1	1.3–5.4	0.02
Grade III	4.3	1.9–12.4	0.009
Surgical margins R2	4.8	2.1–13.7	0.008
**Single analysis (CT)**			
Size	2.9	1.4–8.5	0.01
New Onset Nodule	1.5	1.1–4.1	0.03

Abbreviations: OR = odds ratio; CI = confidence interval.

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
