# Peer review of "Clinical and Radiologic Features Together Better Predict Lung Nodule Malignancy in Patients with Soft-Tissue Sarcoma"

_jcm, 2020, doi:10.3390/jcm9041209_

Round 1

Reviewer 1 Report

The authors described the potential value of the combination of clinical and CT features in the prediction of malignancy of pulmonary nodules in 71 patients with soft tissue sarcoma. The manuscript is very interesting for the originality and the scientific findings. Even though the population seems to be small, it is  absolute acceptable since sarcoma are rare tumors and quality of the final patients selection is very clean.

Minor suggestions:

  1. Considering the value of the findings, I would probably change the title of the manuscript from a question to a statement (but this is not necessary at all);
  2. Regarding chemotherapy, I would report if most of patients received at the least the standard schedule containing antracyclines;
  3. Since the radiologic nature of the contents, I would appreciate one representative figure reporting at least one CT finding (for example with a well defined margins lesion and not), very useful for clinicians

Author Response

Reviewer 1

First of all, we would like to thank this reviewer for the thoughtful comments and efforts towards improving our manuscript

  1. Considering the value of the findings, I would probably change the title of the manuscript from a question to a statement (but this is not necessary at all);

Answer: We agree

Changes: the new title is” Clinical and Radiologic Features Together Better Predict Lung Nodule Malignancy in Patients with Soft Tissue Sarcoma”.

  1. Regarding chemotherapy, I would report if most of patients received at the least the standard schedule containing anthracyclines;

Answer: Chemotherapy fill be the focus of an upcoming paper. Nonetheless, we agree that least this information should be provided to readers.

Changes: Page 6 : “ Forty-six patients (65%) underwent neoadjuvant and/or adjuvant chemotherapy; among them, 40 (87%) received Adriamycin and Ifosfamide.” Was added

  1. Since the radiologic nature of the contents, I would appreciate one representative figure reporting at least one CT finding (for example with a well-defined margins lesion and not), very useful for clinicians

Answer: We agree with this reviewer

Changes: Anew Figure 5 was added

Page 8 “Figure 5 shows some CT findings”. Was added

Figure legends” Figure 5 A. The scan shows a malignant 22 mm round-shaped nodule with well-defined margins (arrow) in the left lower lobe in a 44-year-old female patient with synovial sarcoma. B. The scan shows a 3 mm subpleural round nodule with well-defined margins (arrow) at baseline CT. C. Preoperative CT scan obtained approximately 3 months later in the same patients showing a significant increase in size (11 mm, arrow). The patient received no chemotherapy between the scans. D. The scan shows a 10 mm malignant parenchymal nodule with an irregular shape and spiculation (arrow) in the right lower lobe in a 52-year-old male patient with undifferentiated spindle-cell sarcoma.

Reviewer 2 Report

The article is very relevant, for the correspondence of the diagnostic data, the proposed methodology allows an early identification of the characteristics of benignity or malignancy of lung lesions on CT together with the clinic of STS. Therefore, it highlights a method for a correct therapeutic approach based on the number and size of metastatic lesions. 

The main question addressed by the research is to test the hypothesis that a model including clinical and CT features may allow discrimination between benign and malignant lung nodules in patients with STS.

The question is very relavant and interesting from both statistical and clinical perspectives because the model proposed allow to evaluate the best model in discriminating malignant from benign nodules.

Since Seventy-one patients with STS undergoing first lung metastasectomy were examined, the topic is original and it provides a useful model.

Compared with other published materials, the paper clearly states it key message: neither clinical data nor CT findings alone are able to accurately predict malignant nodules. Conversely, when clinical features are supplemented with CT findings, the predictive value increases significantly, with high sensitivity and specificity. This concept is very helpful because while the primary tumor can be cured with wide surgical resection and a reliable model accurately predicting outcome with different treatment modalities is available, the discrimination between benign and malignant PNs, particularly in small nodules, is challenging.

The paper is well written, clear and easy to read.

The argument is well structured and presented, and conclusions are consistent and well addressed to the main question posed.

Author Response

Reviewer 2

We would like to thank this reviewer. His/her encouraging comments and recognition of our work

is greatly appreciated.

Reviewer 3 Report

I congratulate the authors on a very well written paper and a well conducted retrospective study that clearly delineates its aims and results as well as limitations.

Some of the sentences are very long making it a little difficult to follow what the authors are getting at but overall the grammar is appropriate.

Author Response

Reviewer 3

We would like to thank this reviewer for his / her words of appreciation.

Some of the sentences are very long making it a little difficult to follow what the authors are getting at but overall the grammar is appropriate.

Answer: We agree

Changes: We shortened some long sentences in the discussion and conclusions.